# Better together against genetic heterogeneity: A sex-combined joint main and interaction analysis of 290 quantitative traits in the UK Biobank

Boxi Lin[1], Andrew D. Paterson[1,2]*, Lei Sun[1,3]*

**1** Division of Biostatistics, Dalla Lana School of Public Health, University of Toronto, Toronto, Ontario, Canada, **2** Genetics and Genome Biology, The Hospital for Sick Children, Toronto, Ontario, Canada, **3** Department of Statistical Sciences, University of Toronto, Toronto, Ontario, Canada

* andrew.paterson@sickkids.ca (ADP); lei.sun@utoronto.ca (LS)

**Data Availability Statement:** This research has been conducted using the UK Biobank resource under application number 64875. The sex-stratified GWAS summary statistics were collected from the

## Abstract

Genetic effects can be sex-specific, particularly for traits such as testosterone, a sex hormone. While sex-stratified analysis provides easily interpretable sex-specific effect size estimates, the presence of sex-differences in SNP effect implies a SNP×sex interaction. This suggests the usage of the often overlooked joint test, testing for an SNP's main and SNP×sex interaction effects simultaneously. Notably, even without individual-level data, the joint test statistic can be derived from sex-stratified summary statistics through an omnibus meta-analysis. Utilizing the available sex-stratified summary statistics of the UK Biobank, we performed such omnibus meta-analyses for 290 quantitative traits. Results revealed that this approach is robust to genetic effect heterogeneity and can outperform the traditional sex-stratified or sex-combined main effect-only tests. Therefore, we advocate using the omnibus meta-analysis that captures both the main and interaction effects. Subsequent sex-stratified analysis should be conducted for sex-specific effect size estimation and interpretation.

## Author summary

When genetic variant effects on complex traits differ between females and males, sex-stratified analysis is often applied, offering easy-to-interpret, sex-specific effect estimates. However, from the viewpoint of maximizing the power and robustness of association testing, sex-stratified analysis may not be the best analytical strategy. As sex-specific genetic effects imply an SNP×sex interaction effect, jointly testing SNP main effects and SNP×sex interactions could be more powerful than sex-stratified analysis or the standard main-effect testing. Furthermore, this joint test is applicable even when individual-level data are not available, by leveraging sex-specific summary statistics through an omnibus meta-analysis. In this study, we performed such an omnibus meta-analysis using the UK Biobank data. Across 290 phenotypes, our results showed that this method is generally

Neale Lab: https://github.com/Nealelab/UK_Biobank_GWAS. The GWAS summary statistics to reproduce our analysis can be accessed with DOI 10.5281/zenodo.10836337. Detailed code to reproduce the analysis, including the simulation studies, are provided here: https://github.com/BoxiLin/t2meta.

**Funding:** This work is funded by the Natural Sciences and Engineering Research Council of Canada (NSERC, url: https://www.nserc-crsng.gc.ca/index_eng.asp; grant number: RGPIN-2018-04934 to LS) and the University of Toronto Data Sciences Institute (DSI; url:https://datasciences.utoronto.ca/) Catalyst Grant to ADP and LS. The funders had no role in study design, data collection and analysis, decision to publish, or preparation of the manuscript.

comparable to traditional analyses and excels by identifying new loci for traits such as testosterone, which standard tests do not detect. Our findings suggest that leveraging genetic heterogeneity enhances the detection of genetic associations, with significant implications for the future analysis of diverse data.

## Introduction

The genetic architecture of complex traits can vary among subgroups within a population. Evidence of such heterogeneity has been found in subgroups defined by various factors, including sex [1–4] and smoking status [5]. Sex, in particular, is an important factor, and sex differences in genetic effect on complex traits have been extensively reported [1, 6–8]. Subsequently, the research community has acknowledged the necessity of sex-aware genetic association studies and has developed guidelines for their implementation [3, 9].

A recent genome-wide association study (GWAS) of the UK Biobank data [10] has revealed "small yet widespread" sex differences in genetic architecture across 530 traits [8]. This finding aligns with the observation that genetic effects often exhibit sex-specific patterns, especially for sex hormone traits such as testosterone [1] where sex-stratified GWAS discovered 79 and 127 independent genome-wide significant signals for females and males, respectively, in the UK Biobank [1]. Interestingly, the top GWAS hits for males largely diverged from those for females, both in terms of genomic locations and the signalling pathways in which the annotated genes are involved.

Some earlier GWASs have also reported the existence of genetic effect heterogeneity between sexes. An example of a genetic effect in the same direction but with different magnitudes is serum uric acid concentration [6]: the association between rs7442295 in *SLC2A9* was stronger in females (Effect = -0.46 mg/dl, $p = 2.6 \times 10^{-74}$, explaining 5.8% of the phenotypic variance) than males (Effect = -0.25 mg/dl, $p = 7.0 \times 10^{-17}$, explaining 1.2% of the phenotypic variance). There are also examples of genetic effects in opposite directions. For example, the association between SNPs near *RNF212* and sex-specific recombination rates have opposite directions [7]. Another example is body fat distribution [11–13], for which approximately one-third of all signals identified from sex-stratified GWAS were found to exhibit sex-specific effects [13]. Turning to COVID-19, although there is a well-documented sex difference in the risk of severe outcomes [14], most published GWAS did not conduct sex-stratified analysis or consider SNP×sex interaction with the exception of [15].

In the presence of genetic effect heterogeneity between sexes, a sex-stratified analysis approach is often used, which provides easy-to-interpret, sex-specific *effect size estimates* [16–18]. However, from the viewpoint of maximizing the *power and robustness of association testing*, the sex-stratified analysis may not be the best analytical strategy for two reasons. First, after a sex-stratified analysis, it remains tempting to consider sex-combined analysis by, for example, aggregating the association evidence from both sexes using traditional meta-analysis [19–21], but complications arise. The fixed-effect meta-approach, though it could be used robotically in this context, is conceptually at odds with the heterogeneity setting considered here [22]. Applying the random-effects meta-approach is also challenging, as its implementation requires estimating $\tau^2$, the parameter that quantifies the extent of between-study heterogeneity. When the number of studies (or groups) is small, the estimate can be unreliable [23, 24]. Second, if the effect of an SNP indeed differs between sexes, it implies an SNP×sex interaction effect. This suggests that jointly testing an SNP's main and SNP×sex interaction effects may be more powerful than sex-stratified analysis or the standard main effect-only testing approach [25, 26].

Table 1 summarizes the different types of tests that can be used; the Materials and methods provide more details. Briefly, let $T_{Female}$ and $T_{Male}$ be the summary statistics from sex-stratified analysis, where $T_{Female}$ and $T_{Male}$ follow the standard normal distribution under the null of no association. Now consider three different *meta*-analysis approaches that combine $T_{Female}$ and $T_{Male}$: (1) (for completeness) the traditional meta-analysis $T_{1,metaL}$, using a weighted average of $T_{Female}$ and $T_{Male}$, (2) the SNP×sex interaction effect only test $T_{Diff}$, testing for the effect difference between sexes, and (3) our recommended omnibus meta-analysis $T_{2,metaQ}$, using the sum of $T_{Female}^2$ and $T_{Male}^2$. For the *mega*-analysis of individual data, if available, let $T_{2,mega}$ be the 2 degrees of freedom (df) test derived from jointly testing both the SNP main and SNP×sex interaction effects. For completeness, let us also consider $T_{1,mega}$, the most commonly used GWAS approach of testing the genetic main effect, without the SNP×sex interaction term but with sex included as a covariate in the regression model.

Multiple earlier studies have demonstrated that including interaction effects makes the association analysis more robust to model assumptions in that the joint test remains powerful even in the absence of heterogeneity of SNP effect size between sexes, and the test would be considerably more powerful if there were interactions [25–31]. For example, focusing on gene-environment interaction analysis of a binary trait, Kraft et al [25] concluded that "Although the joint test of genetic marginal effect and interaction is not the most powerful, it is nearly optimal across all penetrance models we considered". As sex can statistically be regarded as an environmental variable, we can infer that the 2 df $T_{2,mega}$ is a more robust test than the 1 df $T_{1,mega}$ in our setting, where SNP×sex is of interest.

When individual data are unavailable, focusing on $T_{1,metaL}$ and $T_{1,mega}$, Lin and Zeng [32] has shown that meta- and mega-analysis perform the same "for all commonly used parametric and semiparametric models." Moreover, Aschard et al [27] also concluded that $T_{2,metaQ}$ and $T_{2,mega}$ are analytically equivalent to each other. Therefore, the 2 df joint main and interaction analysis can be performed by *quadratically* aggregating the sex-stratified summary statistics, where squaring $T_{Female}$ and $T_{Male}$ making the test omnibus; this is akin to the added benefits of SKAT-type over the Burden-type methods when aggregating association evidence across

**Table 1. A list of association methods considered in this analysis.**

| Notation | Method | Details in the UK Biobank (UKB) application | Asymptotic null distribution |
|---|---|---|---|
| Sex-stratified analysis (The UKB sex-stratified analysis was performed by Neale's lab (Online Resources).) | | | |
| $T_{Female}$ | Female-only SNP main effect test | The association summary statistic $T_{Female}$ in females with covariates of 20 principal components, age and age$^2$ | $N(0, 1)$ |
| $T_{Male}$ | Male-only SNP main effect test | The association summary statistic $T_{Male}$ in males with covariates of 20 principal components, age and age$^2$ | $N(0, 1)$ |
| Sex-combined meta-analysis | | | |
| $T_{1,metaL}$ | The traditional meta-analysis (testing an SNP's main effect only) | $\frac{1/\hat{v}_F}{\sqrt{1/\hat{v}_F^2+1/\hat{v}_M^2}} T_{Female} + \frac{1/\hat{v}_M}{\sqrt{1/\hat{v}_F^2+1/\hat{v}_M^2}} T_{Male}$, where $\hat{v}_F$, $\hat{v}_M$ are standard errors of sex-stratified effect estimates | $N(0, 1)$ |
| $T_{Diff}$ | The interaction effect-only test | $\frac{\hat{\beta}_F-\hat{\beta}_M}{\sqrt{\hat{v}_F^2+\hat{v}_M^2}}$, where $\hat{\beta}_F$, $\hat{\beta}_M$, $\hat{v}_F^2$, $\hat{v}_M^2$ are sex-stratified summary statistics | $N(0, 1)$ |
| $T_{2,metaQ}$ | The omnibus meta-analysis (jointly testing an SNP's main and SNP×Sex effects) | $T_{Female}^2 + T_{Male}^2$ | $\chi_2^2$ |
| Sex-combined mega-analysis if individual data are available | | | |
| $T_{1,mega}$ | SNP main effect test | The traditional 1 df test, testing for an SNP's main effect, where sex is included as a covariate | $N(0, 1)$ |
| $T_{2,mega}$ | SNP main and SNP×sex interaction joint analysis | The often overlooked 2 df test, testing for an SNP's main and SNP×sex interaction effects simultaneously | $\chi_2^2$ |

multiple rare variants [33]. The resulting $T_{2,metaQ}$ is more robust to sex-specific effects assumption (e.g. opposite effect direction) than the traditional meta-analysis $T_{1,metaL}$.

While the benefits of the omnibus meta-analysis $T_{2,metaQ}$ (and equivalently the sex-combined joint analysis $T_{2,mega}$ of both main and interaction effects) have been documented, they are yet to be adopted as a standard analysis method for association scanning when heterogeneity is anticipated. Recent genetic research guidelines emphasized the importance of sex-aware genetic association studies [3] and made some general analysis suggestions, including 1) use a blend of two or more strategies of $T_{Female}$, $T_{Male}$, $T_{Diff}$, and $T_{1,mega}$, especially when there is no prior understanding of sex-specific effects, and 2) apply $T_{Diff}$ on a sex-combined sample to verify if the effect size difference is statistically significant, followed by a sex-stratified analysis on a subset of variants with significant interaction [3, 34]. Through a large-scale application, our study aims to provide empirical evidence to show that $T_{2,metaQ}$ is generally comparable to all other GWAS methods and can be more powerful in the presence of genetic effect heterogeneity. Although $T_{2,metaQ}$ method serves as a powerful first-stage association screening method, subsequent stratified analyses, as well as genetic correlation and functional annotation enrichment analyses, should be applied to fully reveal the landscape of genetic effect heterogeneity and its interpretation.

In this analysis, we leveraged publicly available summary statistics from Neale's Lab [35] and conducted a large-scale interaction analysis of 290 complex quantitative traits in the UK Biobank (UKB) data. Our results demonstrate that the omnibus meta-analysis $T_{2,metaQ}$ (equivalently the sex-combined 2 df joint analysis, testing both the main and interaction effects) indeed is more robust to effect heterogeneity assumption than the sex-stratified GWAS ($T_{Female}$ and $T_{Male}$), the traditional meta-analysis ($T_{1,metaL}$ or equivalently $T_{1,mega}$) or testing the SNP-sex interaction effect alone approach ($T_{Diff}$). Focusing on testosterone, a sex-hormone trait with known distinct genetic architectures between sexes, we found that $T_{2,metaQ}$ not only echoed findings from previous GWAS but also unveiled novel signals, suggesting its potential to reveal previously overlooked loci. Moreover, we report multiple serum urate-associated SNPs near *SLC2A9* with genome-wide significant association in both females and males but with opposite effect directions. These signals could be missed by the traditional main effect testing approach ($T_{1,metaL}$ or $T_{1,mega}$), suggesting the importance of a sex-aware approach in future studies.

## Results

Our analysis relied on the sex-stratified summary statistics, labelled as $T_{Female}$ and $T_{Male}$, sourced from the Neale Lab's round 2 (imputed-v3) GWAS [35] (Online Resources), focusing on the 290 continuous traits for which $T_{Female}$ and $T_{Male}$ are available (Materials and methods). The Neale Lab's sex-stratified GWAS included a cohort of up to 361,194 unrelated participants (194,174 females and 167,020 males), genetically determined and self-identified as of white British ancestry; see Online Resources for trait-specific sex-stratified sample sizes. The GWAS originally encompassed 13.7 million genotyped and centrally imputed SNPs, with INFO score > 0.8, minor allele frequency (MAF) > 0.001 (and for SNPs coded as Variant Effect Predictors [36], MAF $> 10^{-6}$), and Hardy-Weinberg equilibrium (HWE) *p*-value $> 10^{-10}$. Considering the varied sample sizes between phenotypes, especially with some as low as approximately 5,000, in our analysis we further focused on SNPs with trait-specific MAF > 0.01 in both females and males. We further assessed the robustness of our results with respect to MAF through sensitivity analyses (Materials and methods).

Using the available sex-stratified summary statistics, $T_{Female}$ and $T_{Male}$, we computed three sex-combined meta-analysis test statistics: the traditional meta-analysis $T_{1,metaL}$, the SNP-sex

interaction-only $T_{Diff}$, and the omnibus meta-analysis $T_{2,metaQ}$ (Table 1; Materials and methods). We note $T_{1,mega}$ and $T_{2,mega}$ were omitted from this analysis, as they are empirically equivalent to, respectively, $T_{1,metaL}$ and $T_{2,metaQ}$ (Fig i in S1 Appendix), consistent with the analytical conclusion of [32] and [26]. We used the threshold of p-value $< 5 \times 10^{-8}$ to declare genome-wide significance [37].

## Genome-wide comparison of sex-stratified and sex-combined association methods across traits

There are 1,113,865 SNPs genome-wide significantly associated with one or more traits in the 290 traits analyzed, detected by any of the five association testing methods. Fig 1A shows the pairwise scatter plots for the corresponding set of 6,207,519 genome-wide significant SNP-phenotype associations. The first notable feature is that the $-log_{10}$ p-values of $T_{2,metaQ}$ appear to be consistently larger than those of the other four methods (the last column in Fig 1A), indicating that $T_{2,metaQ}$ is capable of capturing signals detected by any of the other methods.

The second feature in Fig 1A, when comparing $T_{1,metaL}$ with $T_{2,metaQ}$, is the presence of a cluster of SNPs close to the x-axis, where the $-log_{10}$ p-values of $T_{1,metaL}$ are close to zero but the $-log_{10}$ p-values of $T_{2,metaQ}$ are very large, indicating substantially better performance by $T_{2,metaQ}$ as compared with $T_{1,metaL}$. Upon further inspection, these associations are primarily for testosterone (Fig 1B). After removing the testosterone, Fig 1B shows that the first feature remains: For all significant associations identified by any of the five methods, association evidence provided by $T_{2,metaQ}$ are comparable or substantially stronger. In other words, $T_{2,metaQ}$ is a robust association method. We arrived at similar conclusions from MAF-stratified pairwise scatter plots (MAF $> 0.05$ in both sex groups vs. either sex-stratified MAF $\leq 0.05$; S1 Fig). Alternatively, focusing on traits, instead of SNPs, we compared the the number of associated traits across 1,113,865 SNPs among the five methods (S2–S4 Figs), and concluded that method comparison remain consistent.

Fig 2A shows the stacked-Manhattan plots of the set of 179,718 genome-wide significant SNPs identified by the recommended omnibus meta-analysis $T_{2,metaQ}$ but missed by the traditional meta-analysis $T_{1,metaL}$ across the 290 traits analyzed. It is evident that SNPs identified by $T_{2,metaQ}$ have $-log_{10}$ p-value range from 7.3 to over 200, but they may have no association evidence based on $T_{1,metaL}$ with $-log_{10}$ p-value close to zero. On the other hand, although the number of SNPs missed by $T_{2,metaQ}$ but identified by $T_{1,metaL}$ is 756,316, sustainably larger than 179,718, the two sets of p-values among the 756,316 SNPs are comparable (Fig 2B; also see S5 Fig for distribution of these SNPs with a zoomed-in view within Fig 1A). Notably, the minimum $-log_{10}(p)$ for such SNPs was 6.45 by $T_{2,metaQ}$ while capped at 8.17 by $T_{1,metaL}$, matching the expected small cost of $T_{2,metaQ}$ having an additional 1 degree of freedom in the absence of sex difference in genetic effect (S7 Appendix). This pattern of results persisted, albeit to a lesser degree, after removing testosterone (Fig 2C and 2D). Further, a similar conclusion can be drawn when $T_{2,metaQ}$ is compared with the aggregated SNPs identified by any of the alternative methods considered (S6 Fig) or based on MAF-stratified results (S7–S10 Figs).

Fig 3A contrasts the numbers of genome-wide significant SNPs identified by either $T_{1,metaL}$ or $T_{2,metaQ}$ for each of the 290 traits analyzed; see S11 Fig for a $log_{10}$ transformation, which allows better visualization of the traits with smaller numbers of associated SNP. Although it is clear that $T_{1,metaL}$ generally resulted in more associated SNPs than $T_{2,metaQ}$, it is important to recall that, for SNPs missed by $T_{2,metaQ}$, $T_{2,metaQ}$ provides *comparable* association evidence (the last two rows in Fig 2B and 2D). On the other hand, for SNPs missed by $T_{1,metaL}$, $T_{1,metaL}$ may provide no association evidence at all (the last two rows in Fig 2A and 2C), completely failing to capture those SNPs identified by $T_{2,metaQ}$. Both features highlight the fact that $T_{2,metaQ}$ is

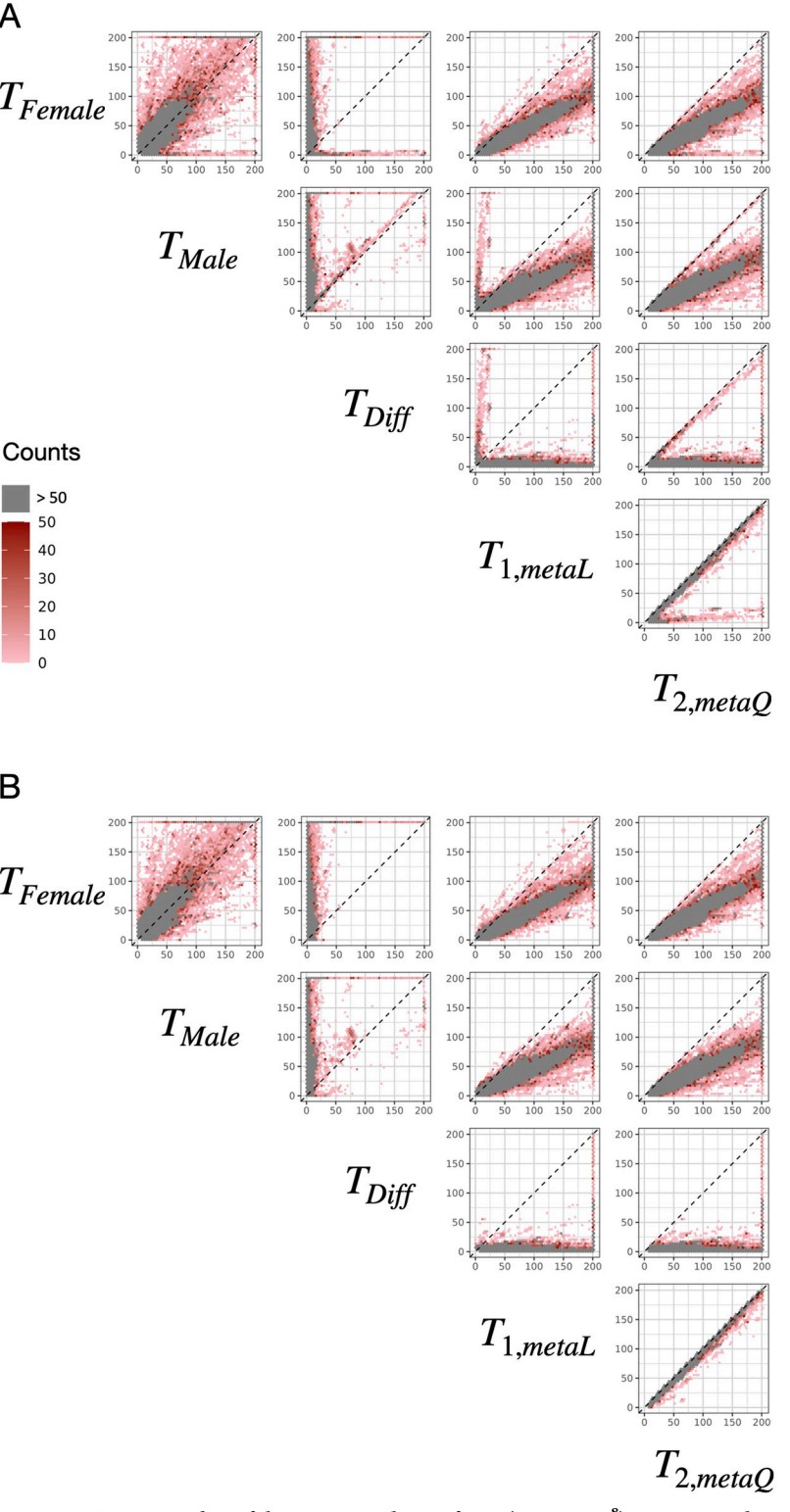

**Fig 1. Pairwise scatter plots of the genome-wide significant ($p < 5 \times 10^{-8}$) associations detected by any of the five testing methods considered.** (A) includes 6,207,519 SNP-phenotype associations across all 290 traits analyzed, and (B) includes 6,175,594 SNP-phenotype associations after excluding testosterone. The five association methods include $T_{Female}$ (Female-only analysis), $T_{Male}$ (Male-only analysis), $T_{Diff}$ (SNP-sex interaction-only test), $T_{1,metaL}$ (the traditional sex-combined meta-analysis), and $T_{2,metaQ}$ (the omnibus meta-analysis); see Table 1 for method details. The sex-

stratified GWAS summary statistics come from the Neale lab's UK Biobank GWAS round 2, which included a cohort of up to 361,194 participants (194,174 females and 167,020 males). Axes depict $-\log_{10}$ p-values for each pair of tests, and each hexagon's color corresponds to the count of associations falling within the $-\log_{10}$ p-value range defined by that region. Dashed line indicates the main diagonal reference line. The $-\log_{10} p$ maximum was truncated at 200 to improve visualization.

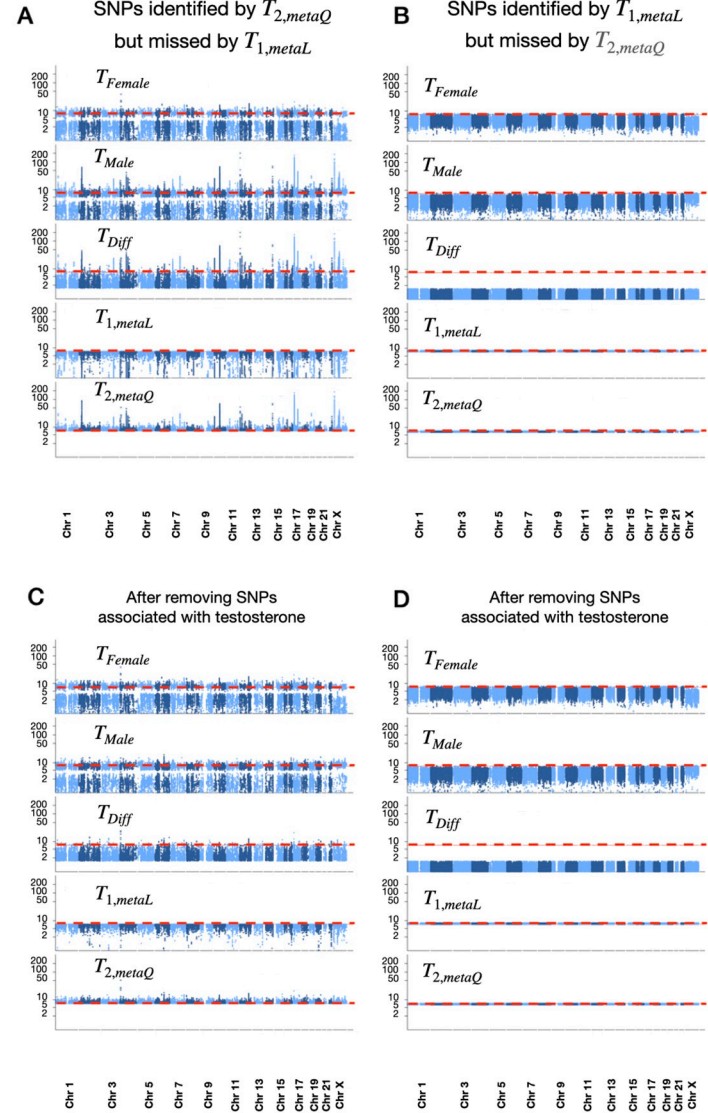

**Fig 2. Stacked Manhattan plots of genome-wide significant SNPs.** (A) 179,718 SNP-phenotype associations identified by $T_{2,metaQ}$ but missed by $T_{1,metaL}$ across all the 290 traits, (B) 756,316 SNP-phenotype associations missed by $T_{2,metaQ}$ but identified by $T_{1,metaL}$ across all the 290 traits, (C) 155,930 SNP-phenotype associations identified by $T_{2,metaQ}$ but missed by $T_{1,metaL}$ after removing the testosterone, and (D) 756,051 SNP-phenotype associations missed by $T_{2,metaQ}$ but identified by $T_{1,metaL}$ after removing the testosterone. The $-\log_{10}$ p-values (with further $\log_{10}$ transformation on y-axis to aid presentation) are shown for the five association methods, including $T_{Female}$ (Female-only analysis), $T_{Male}$ (Male-only analysis), $T_{Diff}$ (SNP-sex interaction-only test), $T_{1,metaL}$ (the traditional sex-combined meta-analysis), and $T_{2,metaQ}$ (the omnibus meta-analysis); see Table 1 for method details. The sex-stratified GWAS summary statistics come from the Neale lab's UK Biobank GWAS round 2, which included a cohort of up to 361,194 participants (194,174 females and 167,020 males). The red horizontal lines (at $y = -\log_{10}(5 \times 10^{-8}) = 7.3$) indicate the genome-wide significant threshold of $5 \times 10^{-8}$ on the $-log_{10}$ scale. See S6 Fig for similar stacked Manhattan plots when comparing SNPs identified by $T_{2,metaQ}$ with those identified by any of the alternative methods considered.

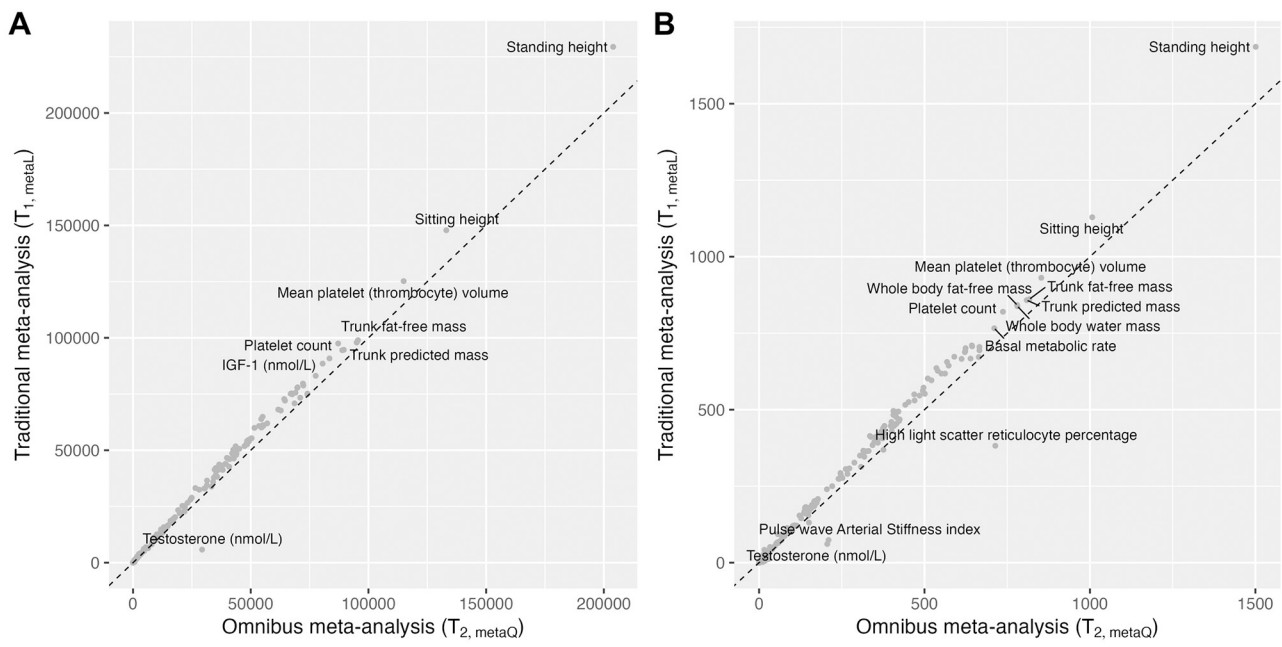

**Fig 3. Comparison of the numbers of genome-wide significant (A) SNPs and (B) independent loci, identified by the traditional meta-analysis** $T_{1,metaL}$ **(y-axis) and the omnibus meta-analysis** $T_{2,metaQ}$ **(x-axis) for each of the 290 traits analyzed.** The dashed line indicates the reference main diagonal line. See S11 Fig for a log10 transformation of the axes to better visualization traits with smaller numbers of associated SNPs. See S12 Fig for a zoomed-in plot focusing on traits with smaller numbers of associated loci, including testosterone, high light scatter reticulocyte percentage and pulse wave arterial stiffness index; see S4 Appendix for additional analyses of these phenotypes. In (B), the linkage disequilibrium is defined with a physical distance of 10MB and $r^2 > 0.01$ (Materials and methods); see Fig i in S4 Appendix for the loci results after using a different linkage disequilibrium threshold.

more robust than $T_{1,metaL}$ against genetic heterogeneity assumption. These features are consistent when comparing independent SNPs after LD clumping (Materials and methods; Fig 3B), albeit with four phenotypes, in addition to testosterone, showing more significant loci in $T_{2,metaQ}$, for which we provided a comprehensive analysis check in S4 Appendix. A graphical comparison of the different methods for each of the 290 traits can be found at https://sex-combined-interaction-meta.shinyapps.io/figures/.

When genetic heterogeneity is present, such as for testosterone (UKB Data-Field: 30850), $T_{2,metaQ}$ is substantially more powerful than $T_{1,metaL}$, which we examine closely next.

## Genome-wide results for testosterone

Noticeably, testosterone (UKB Data-Field: 30850) behaves differently from other traits, where the omnibus meta-analysis $T_{2,metaQ}$ yielded substantially more genome-wide significant SNPs than the traditional meta-analysis $T_{1,metaL}$ (29,327 and 5,804 for $T_{2,metaQ}$ and $T_{1,metaL}$ respectively, corresponding to 206 and 61 independent loci; Materials and methods). Given the established distinct genetic architecture of testosterone levels between males and females [1], this observation suggests that association analysis of phenotypes with sex-specific genetic effects could benefit considerably from the use of omnibus meta-analysis. Indeed, the 2 df $T_{2,metaQ}$ test generally recovers all hits detected by all of the other four 1 df methods (S14(A) Fig). Additionally, the $-\log_{10}(p)$ values derived from $T_{2,metaQ}$ are always nearly equivalent to, or significantly larger than, those from the other tests (S14(B) Fig). This observation is similar to that drawn from evaluating all traits (Fig 1), suggesting that $T_{2,metaQ}$ exhibits enhanced robustness and potentially superior power relative to the other testing methods. Notably, many SNPs

identified by $T_{2,metaQ}$ but overlooked by other methods exhibit opposite sex-specific effect directions between females and males (S14(B) Fig).

For testosterone, $T_{2,metaQ}$ uniquely identified 1,287 genome-wide significant SNPs which corresponds to 36 independent loci (Materials and methods). We ascertained the biological relevance of the 36 loci by cross-referencing the NHGRI-EBI GWAS catalog [38] (Version: e0r2022-11-29) to check for previously reported SNP-phenotype associations of the lead SNPs and nearby SNPs in linkage disequilibrium (LD; Materials and methods). We found four of these 36 loci include SNPs with reported association with testosterone in two recent GWAS [39, 40] (S1(A) Table). Even though these loci are not novel, our confirmation of these associations using the Neale Lab's UK Biobank data underscores the utilizes of $T_{2,metaQ}$, since these loci are overlooked by either the sex-stratified approach or the conventional sex-combined methods.

Of the remaining 32 loci, 12 included lead SNPs (or SNPs in LD) with previously reported associations with other traits (S1(B) Table). Although these SNPs have no previously reported association with testosterone levels directly, some (e.g. sex hormone-binding globulin levels) are related to testosterone levels and to the steroid biosynthesis pathway—the process responsible for testosterone synthesis.

Moreover, joint testing through $T_{2,metaQ}$ also revealed 20 novel loci with no association reported in the NHGRI-EBI GWAS catalog (date accessed: 2023-05-13). These newly discovered loci merit further investigation. Table 2 lists results for the 12 loci with bi-allelic leading SNPs; S2 Table shows results for the remaining 8 loci with leading insertions/deletions (INDELs). Interestingly, the features of the 12 leading bi-allelic SNPs in Table 2 are similar to those 4 SNPs in S1(A) Table (with earlier association reports in the NHGRI-EBI GWAS

**Table 2. A total of 12 novel testosterone-associated loci with leading bi-allelic SNPs uniquely identified by the recommended $T_{2,metaQ}$ but missed by any other methods in the UK Biobank data application.**

| Lead SNP | CHR | BP (hg19) | INFO | Major / Minor Allele | MAF (All/Female/Male) | $\beta_{Female}$ | $\beta_{Male}$ | $P_{Female}$ | $P_{Male}$ | $P_{Diff}$ | $P_{1,metaL}$ | $P_{2,metaQ}$ |
|---|---|---|---|---|---|---|---|---|---|---|---|---|
| rs4308991 | 1 | 93418330 | 0.987 | T / A | (0.398 / 0.398 / 0.397) | -0.007 | 0.068 | 2.01E-03 | 5.73E-07 | 5.20E-08 | 2.69E-02 | 3.16E-08 |
| rs2638052 | 1 | 101956313 | 0.999 | A / T | (0.340 / 0.341 / 0.339) | -0.01 | 0.059 | 9.95E-06 | 2.45E-05 | 9.76E-07 | 2.58E-04 | 7.84E-09 |
| rs72697614 | 1 | 107514107 | 0.972 | A / C | (0.321 / 0.320 / 0.322) | 0.008 | 0.078 | 8.90E-04 | 5.06E-08 | 1.44E-06 | 2.85E-05 | 1.42E-09 |
| rs1672939 | 3 | 138276133 | 0.999 | T / C | (0.449 / 0.449 / 0.449) | 0.006 | 0.072 | 5.37E-03 | 6.31E-08 | 1.11E-06 | 2.67E-04 | 9.15E-09 |
| rs150750289 | 4 | 3443769 | 0.950 | T / C | (0.024 / 0.024 / 0.024) | -0.025 | -0.209 | 7.52E-04 | 1.83E-06 | 3.45E-05 | 3.78E-05 | 3.88E-08 |
| rs4946386 | 6 | 119322992 | 0.984 | T / C | (0.302 / 0.302 / 0.302) | 0.013 | -0.037 | 1.91E-07 | 1.08E-02 | 7.28E-04 | 2.44E-06 | 4.99E-08 |
| rs117916257 | 10 | 65387009 | 0.976 | G / A | (0.044 / 0.044 / 0.044) | -0.015 | -0.176 | 7.17E-03 | 5.69E-08 | 9.26E-07 | 3.81E-04 | 1.07E-08 |
| rs7943570 | 11 | 48085968 | 0.994 | T / C | (0.090 / 0.090 / 0.090) | -0.009 | -0.125 | 2.40E-02 | 6.86E-08 | 7.68E-07 | 1.78E-03 | 3.74E-08 |
| rs2241235 | 17 | 7381366 | 0.982 | C / T | (0.141 / 0.140 / 0.142) | -0.009 | -0.099 | 8.08E-03 | 1.87E-07 | 2.65E-06 | 4.96E-04 | 3.77E-08 |
| rs8087735 | 18 | 71977923 | 0.979 | G / A | (0.140 / 0.140 / 0.139) | 0.017 | -0.045 | 7.45E-08 | 1.96E-02 | 1.39E-03 | 8.80E-07 | 3.39E-08 |
| rs117599084 | 19 | 49530502 | 0.890 | A / T | (0.073 / 0.074 / 0.073) | 0.018 | 0.114 | 8.15E-05 | 1.91E-05 | 3.72E-04 | 4.29E-06 | 4.58E-08 |
| rs146539762 | 23 | 65720145 | 0.943 | A / G | (0.017 / 0.017 / 0.017) | -0.021 | -0.196 | 1.91E-02 | 9.98E-08 | 3.57E-06 | 4.33E-04 | 4.42E-08 |

$T_{2,metaQ}$ uniquely identified 1,287 genome-wide significant testosterone-associated SNPs that were missed by any of the four alternative methods, encompassed by 36 independent loci after LD clumping. This table presents 12 of these 36 loci with no reported association in the NHGRI-EBI GWAS catalog [38] (Version: e0r2022-11-29). Each locus is represented by its lead SNP which has the smallest $T_{2,metaQ}$ p-value within the locus. The table only included loci with bi-allelic leading SNPs; loci with leading insertions/deletions (INDELs) are reported in S2 Table. The p-values are from the following methods: $P_{Female}$: Female-only analysis ($T_{Female}$); $P_{Male}$: Male-only analysis ($T_{Male}$), $P_{Diff}$: SNP-sex interaction-only test ($T_{Diff}$), $P_{1,metaL}$: Traditional sex-combined meta-analysis ($T_{1,metaL}$), and $P_{2,metaQ}$: Omnibus meta-analysis ($T_{2,metaQ}$). The $\beta_{Female}$ and $\beta_{Male}$ columns show the sex-specific effect size estimates from the stratified analysis, indicating the estimated effect of each copy of the minor allele. The sex-stratified GWAS summary statistics came from the Neale lab's UK Biobank GWAS round 2, which included a cohort of up to 361,194 participants (312,102 for the testosterone GWAS, 154,364 females and 157,738 males). Notably, testosterone exhibits a marked difference in both phenotype mean and variance between sexes (Fig iii (A) in S4 Appendix).

catalog), including similar sex-stratified MAF and notable (but not significant) association evidence in one sex. This consistency in results between S1(A) Table and Table 2 further supports the use of $T_{2,metaQ}$.

Finally, out of the 290 phenotypes analyzed, 164 phenotypes have one or more associated loci that were identified exclusively by the $T_{2,metaQ}$ and not by any of the other four 1 d.f. tests. The distribution for these loci numbers is illustrated in S13 Fig, and detailed information about the index SNPs of these loci are available in S1 Data.

### Variants with sex-divergent genetic effects on serum urate levels

Of all GWAS signals across all 290 continuous traits analyzed, 19 bi-allelic SNPs showed genome-wide significant effects on serum urate levels (UKB Data-Field: 30880; Fig i in S6 Appendix) with opposite directions between sexes (Table 3); all 19 SNPs are located near *SLC2A9* on Chr4:10004389-10049700 (Fig ii-Fig vi in S6 Appendix) and in strong LD with each other ($r^2 \approx 1$). The 19 SNPs are independent of rs7442295 previously reported by Döring et al [6] ($r^2 = 0.004$ between rs7442295 and rs6857001, the leading SNP of the 19 SNPs with smallest $T_{2,metaQ}$ p-value). Additionally, although rs7442295 exhibited different effect magnitudes between sexes, the directions are consistent [6]. In contrast, the novel SNPs identified in our study have sex-specific directions: The minor alleles are associated with lower serum urate levels in females but higher levels in males; the effect magnitudes also differ between the sexes.

Aligned with our analytical expectation, $T_{1,metaL}$ exhibited substantially reduced statistical significance as compared to $T_{2,metaQ}$. Two SNPs (rs6814556 and rs6833878) did not even reach

**Table 3. A total of 19 bi-allelic SNPs exhibiting genome-wide significant effects in opposite directions between sexes, all associated with urate levels in the UK Biobank data application.**

| SNP | CHR | BP (hg19) | INFO | Major / Minor Allele | MAF (All/Female/Male) | $\beta_{Female}$ | $\beta_{Male}$ | $P_{Female}$ | $P_{Male}$ | $P_{Diff}$ | $P_{1,metaL}$ | $P_{2,metaQ}$ |
|---|---|---|---|---|---|---|---|---|---|---|---|---|
| rs6814556 | 4 | 10004389 | 0.999 | A / G | (0.268 / 0.268 / 0.268) | -3.184 | 2.285 | 4.91E-40 | 1.44E-15 | 1.77E-48 | 5.37E-07 | 1.14E-52 |
| rs6833878 | 4 | 10005555 | 1.000 | A / T | (0.268 / 0.268 / 0.268) | -3.169 | 2.277 | 1.09E-39 | 1.80E-15 | 4.25E-48 | 6.25E-07 | 3.14E-52 |
| rs3796834 | 4 | 10012846 | 1.000 | C / T | (0.279 / 0.279 / 0.279) | -3.704 | 2.056 | 8.14E-55 | 3.47E-13 | 6.77E-55 | 3.86E-13 | 4.73E-65 |
| rs3796833 | 4 | 10012878 | 1.000 | C / T | (0.279 / 0.279 / 0.279) | -3.714 | 2.052 | 4.33E-55 | 3.86E-13 | 5.31E-55 | 2.88E-13 | 2.81E-65 |
| rs1122966 | 4 | 10014476 | 1.000 | G / A | (0.280 / 0.279 / 0.280) | -3.813 | 2.002 | 4.36E-58 | 1.32E-12 | 4.90E-56 | 1.00E-14 | 9.63E-68 |
| rs6857001 | 4 | 10018080 | 1.000 | G / A | (0.280 / 0.280 / 0.280) | -3.837 | 1.980 | 7.88E-59 | 2.24E-12 | 4.06E-56 | 3.63E-15 | 2.93E-68 |
| rs3796830 | 4 | 10019135 | 1.000 | G / C | (0.279 / 0.279 / 0.279) | -3.742 | 2.033 | 6.43E-56 | 6.18E-13 | 3.38E-55 | 1.05E-13 | 6.58E-66 |
| rs28449404 | 4 | 10019678 | 1.000 | C / G | (0.279 / 0.279 / 0.279) | -3.738 | 2.033 | 8.34E-56 | 6.21E-13 | 4.03E-55 | 1.15E-13 | 8.64E-66 |
| rs10030570 | 4 | 10027160 | 1.000 | G / T | (0.280 / 0.280 / 0.280) | -3.687 | 2.100 | 1.79E-54 | 9.98E-14 | 1.51E-55 | 1.12E-12 | 3.03E-65 |
| rs6819833 | 4 | 10027354 | 1.000 | C / G | (0.280 / 0.280 / 0.280) | -3.687 | 2.100 | 1.85E-54 | 1.01E-13 | 1.55E-55 | 1.13E-12 | 3.14E-65 |
| rs6820230 | 4 | 10027542 | 1.000 | C / T | (0.280 / 0.280 / 0.280) | -3.691 | 2.095 | 1.41E-54 | 1.15E-13 | 1.62E-55 | 9.45E-13 | 2.76E-65 |
| rs6449237 | 4 | 10027643 | 1.000 | A / G | (0.280 / 0.280 / 0.281) | -3.689 | 2.095 | 1.60E-54 | 1.15E-13 | 1.77E-55 | 9.86E-13 | 3.13E-65 |
| rs6449238 | 4 | 10027744 | 1.000 | G / A | (0.280 / 0.280 / 0.280) | -3.689 | 2.100 | 1.60E-54 | 1.01E-13 | 1.43E-55 | 1.07E-12 | 2.75E-65 |
| rs7697004 | 4 | 10028077 | 1.000 | G / A | (0.280 / 0.280 / 0.281) | -3.686 | 2.101 | 1.95E-54 | 9.77E-14 | 1.54E-55 | 1.18E-12 | 3.25E-65 |
| rs7697416 | 4 | 10028287 | 0.999 | G / A | (0.281 / 0.280 / 0.281) | -3.659 | 2.105 | 1.13E-53 | 8.62E-14 | 3.95E-55 | 2.38E-12 | 1.66E-64 |
| rs7669699 | 4 | 10028438 | 1.000 | C / T | (0.280 / 0.280 / 0.280) | -3.693 | 2.097 | 1.24E-54 | 1.10E-13 | 1.38E-55 | 9.32E-13 | 2.32E-65 |
| rs9291645 | 4 | 10038254 | 0.997 | G / A | (0.281 / 0.281 / 0.282) | -3.654 | 2.145 | 1.43E-53 | 2.97E-14 | 9.00E-56 | 4.65E-12 | 7.31E-65 |
| rs6850166 | 4 | 10043688 | 0.997 | C / T | (0.281 / 0.281 / 0.281) | -3.709 | 2.115 | 4.09E-55 | 6.94E-14 | 3.38E-56 | 7.99E-13 | 4.89E-66 |
| rs4391034 | 4 | 10049700 | 0.996 | T / C | (0.282 / 0.281 / 0.282) | -3.688 | 2.105 | 1.59E-54 | 9.03E-14 | 1.25E-55 | 1.10E-12 | 2.46E-65 |

The p-values are from the following methods: $P_{Female}$: Female-only analysis ($T_{Female}$); $P_{Male}$: Male-only analysis ($T_{Male}$), $P_{Diff}$: SNP-sex interaction-only test ($T_{Diff}$), $P_{1,metaL}$: Traditional sex-combined meta-analysis ($T_{1,metaL}$), and $P_{2,metaQ}$: Omnibus meta-analysis ($T_{2,metaQ}$). The $\beta_{Female}$ and $\beta_{Male}$ columns show the sex-specific effect size estimates from the stratified analysis, indicating the estimated effect of each copy of the minor allele. The sex-stratified GWAS summary statistics came from the Neale lab's UK Biobank GWAS round 2, which included a cohort of up to 361,194 participants (343,836 in urate GWAS, 184,755 females and 159,081 males).

genome-wide significance of $<5 \times 10^{-8}$ based on $T_{1,metaL}$, while the p-values of $T_{2,metaQ}$ are $<5 \times 10^{-50}$. We cross-referenced our results with those from a recent GWAS meta-analysis of serum urate [41] but did not find any of these 19 SNPs on their list of SNPs showing sex-specific differences (Table 9 of [41]), nor in their credible sets of SNPs with 99% posterior probability of containing the variant(s) driving four independent association signals at this locus (Table 18 of [41]). This is, however, unsurprising as the meta-analysis of [41] was based on $T_{1,metaL}$ which may miss SNPs with sex-specific directions, while our novel discoveries are based on the omnibus $T_{2,metaQ}$.

We further investigated the robustness of our conclusion based on the summary statistics of these 19 SNPs from the GWAS of inverse normal transformed (IRNT) urate. We observed consistent results with those based on the original, un-transformed urate values in terms of both effect direction and relative performance between the different tests (S3 Table).

## Discussion

Often, heterogeneity is viewed as a challenge that may lead to decreased statistical power. However, we illustrate in this study that genetic effect heterogeneity can be exploited to enhance statistical power to detect genetic associations. Notably, such an advantage can be achieved through straightforward calculations when sex-stratified summary statistics are readily available.

By leveraging the publicly available sex-stratified GWAS results from Neale's lab [35] (http://www.nealelab.is/uk-biobank), we undertook a comprehensive comparison of sex-stratified and sex-combined analysis methodologies across 290 quantitative complex traits from the UK Biobank. Our findings underscore that while sex-combined 2 df interaction analysis is generally comparable to sex-stratified GWAS, the traditional meta-analysis on main effect only test ($T_{1,metaL}$), or SNP-sex interaction effect only test ($T_{Diff}$), it can surpass all these methods under genetic effect heterogeneity. Although we only analyzed continuous phenotypes, our findings and conclusions can be extended to binary and ordinal outcomes provided that the sample size is sufficient [42] and extreme case-control, if present, is addressed [43].

We narrowed our focus to serum testosterone, a sex-hormone trait, known to possess distinct genetic architectures between sexes. Intriguingly, the unique signals detected by $T_{2,metaQ}$ not only echo findings recorded in the NHGRI-EBI GWAS catalogue resource [38], but also reveal novel signals not previously recorded in the same resource. These results suggest that $T_{2,metaQ}$ can potentially unearth signals that previous studies may have overlooked.

Furthermore, we highlighted the presence of multiple serum urate-associated SNPs near *SLC2A9* with genome-wide significant associations in both females and males, but with contrasting effect directions. These associations may not be detected when employing the conventional meta-analysis approach ($T_{1metaL}$), underscoring the critical need for the integration of sex-aware methodologies in future investigations. Noticeably, the *SLC2A9* region harbours at least four independent signals associated with serum urate [6, 41].

Certainly, the relative power between these tests is contingent on the underlying genetic models, and may vary considerably across different scenarios. For instance, $T_{1,mega}$, derived from the standard genetic main effect model, is optimal when the true genetic effects exhibit homogeneity between males and females. However, when the effects display opposite directions with similar magnitudes, $T_{1,mega}$ loses its capacity to detect association. In contrast, the female-only sex-stratified test, $T_{Female}$, is optimal when the effect is exclusive to females, but its relative power to sex-combined methods may diminish substantially when the effect also manifests in males. Although $T_{Diff}$ is commonly utilized for testing sex-dimorphic SNP (sdSNP

[8]), its relative deficiency in power for testing SNP-sex interaction effects, even when compared with $T_{1,mega}$, has been illuminated in analytical studies [26].

To provide additional supporting evidence to the existing literature [25–31, 44], we conducted extensive simulation studies to show the robustness of jointly testing both SNP main effect and SNP-sex interaction effect, including adequate type I error control (S2 Appendix and Fig ii-Fig x in S3 Appendix). $T_{2,mega}$ (and equivalently $T_{2,metaQ}$) retains power even in the absence of SNP-sex interaction or if the effect is solely present in one group. For each scenario considered, $T_{2,mega}$ and $T_{2,metaQ}$ are either the most powerful test or experience only minor power loss compared with the oracle method tailored to that specific scenario.

We acknowledge certain limitations in our study. Firstly, as our analysis includes all 290 continuous raw phenotypes available from Neale's Lab, we utilized the results from their linear regression model for all evaluated traits, without conducting an explicit examination of the model assumptions and sample quality control for each individual trait. Therefore, the effect estimates should be interpreted with caution. For example, of the 194,174 females passed quality control, near 20% were excluded from the testosterone GWAS because their measurements fell below the lower detection limit (0.35 nmol/L). In comparison, in the 167,020 males, only 0.01% were removed due to measurements beneath lower detection limit (0.35 nmol/L) and 0.001% were excluded for exceeding the upper detection limit (55 nmol/L). Such unbalanced truncation of lower and upper extreme values could potentially introduce participation bias [45]. Statistical models such as Tobit model [46] could be applied to incorporate data truncation.

Additionally, our analysis focused on GWAS of raw phenotypes instead of rank-based inverse normal transformation (IRNT). Association studies with raw phenotypes may be biased by outlier samples with extreme phenotypic values. Indeed, in comparing the number of independent SNPs after LD clumping (Fig 3B), we observed that $T_{2,metaQ}$ identified excess of loci over $T_{1,metaL}$ for four phenotypes in addition to testosterone. Such excess was not observed after applying IRNT, highlighting potential false discovery from extreme outliers or non-normality in residuals (Fig ii-Fig ix in S4 Appendix). We recommend conducting thorough sensitivity analyses in future research to determine if extreme phenotypic values arise from genuine genetic (or environmental) factors or measurement errors.

Although association tests with IRNT phenotypes may mitigate type I errors, sex-specific transformations (used by Neale's lab for the UKB data), unfortunately, lead to inconsistent response variables (quantiles) between stratified and sex-combined samples. This inconsistency disrupts the equivalence between meta- and mega-analysis, especially for phenotypes exhibiting mean and variance differences between sexes (Fig iii(A) in S4 Appendix), such as testosterone, therefore complicates interpretation. Alternative strategies, such as *log*-transformation [1], winsorization or removal of extreme values [47], could offer protection against extreme outliers without introducing inconsistency in the response variables across different sample groupings.

Moreover, although our analyses were restricted to unrelated individuals [35], it is crucial to approach the implementation of $T_{2,metaQ}$ cautiously if sample relatedness is present. Firstly, the presence of related samples between sexes compromises the independence of $T_{Female}$ and $T_{Male}$, disrupting the expected $\chi_2^2$ distribution of $T_{2,metaQ}$ under the null. This situation is akin to the challenges in meta-analyzing overlapping samples in the context of $T_{1,metaL}$, where neglecting sample relatedness may increase type I errors [48]. Secondly, within-sex sample relatedness, while it could be readily accounted for in sex-stratified analyses, raises questions about the analytical equivalence between $T_{2,metaQ}$ and $T_{2,mega}$. Thus, addressing these between- and within-sex relatedness in the context of $T_{2,metaQ}$ is of future research interest.

Finally, while our analysis encompassed SNPs from both autosomes and X-chromosomes, we did not specifically address the analytical challenges and nuances associated with the X-chromosome [49, 50]. X-chromosome specific mechanisms, such as X-chromosome dosage compensation, may contribute to X-linked sex-specific effect heterogeneity [51]. Consistent with earlier analysis, we confirmed through our study, rs5934505 on X-chromosome showed male-only genome-wide significant effects in 38 out of the 290 continuous phenotypes analyzed (S15 Fig). Future directions would involve conducting a more refined analysis and methodological comparison specifically tailored to the analysis of the sex chromosomes.

In conclusion, our findings strongly support the utilization of $T_{2,mega}$ (or its equivalent omnibus meta-analysis $T_{2,metaQ}$ when sex-stratified summary statistics are available) for the initial screening of genetic associations, particularly in scenarios where genetic heterogeneity is anticipated. We further recommend follow-up sex-stratified analysis, to elucidate the complete landscape of genetic effect heterogeneity and interpretation.

## Materials and methods

### Data overview

Our analysis is based on sex-stratified GWAS summary statistics sourced from the Neale Lab's round 2 (imputed-v3) GWAS [35], which included a cohort of up to 361,194 unrelated participants (194,174 females and 167,020 males), genetically determined and self-identified as of white British ancestry. Additional inclusion criteria for the sample encompassed inclusion in principal component calculations and sex chromosome euploidy.

### Phenotypes

The Neale Lab's round 2 GWAS analyzed 4,203 distinct phenotypes in the UK Biobank. As the primary goal of our analysis is to compare methodologies, we opted to focus on continuous phenotypes and excluded all binary and ordinal phenotypes to avoid the potential additional complexities caused by inadequate statistical power [8, 47] or extreme case-control imbalance [43]. We further narrowed our scope to the raw measures of continuous phenotypes without rank-based inverse normal transformation (IRNT). Finally, we excluded 15 phenotypes where any of the sex-stratified or sex-combined summary statistics were not available, resulting in a total of 290 continuous traits analyzed in our study. For each of these phenotypes, our analysis only used the available female and male GWAS summary statistics.

### Marker's quality control

Neale Lab's GWAS encompassed 13.7 million genotyped and centrally imputed SNPs on autosomes and X chromosome, with INFO score > 0.8, minor allele frequency (MAF) > 0.001 (and for SNPs coded as Variant Effect Predictors [36], MAF $> 10^{-6}$), and Hardy-Weinberg equilibrium (HWE) $p$-value $> 10^{-10}$. In our analysis, given the variability in sample sizes across phenotypes, some as small as approximately 5,000, we specifically focused on SNPs that have a trait-specific MAF greater than 0.01 in both females and males.

### Association testing

In this section we provide details for the sex-stratified, sex-combined meta- and mega-analysis considered in this study.

**$T_{Female}$ and $T_{Male}$, the female- and male-only SNP main effect tests.**  Let $T_{Female} = \hat{\beta}_F / \hat{v}_F$ be the standard Wald test statistic derived from the female-only regression model, where $\hat{\beta}_F$ and $\hat{v}_F$ represent the genetic effect estimates and their standard errors using female samples,

respectively. In Neale Lab's sex-stratified GWAS, covariates such as the first 20 principal components, age, and age squared were controlled. Similarly, we can obtain $T_{Male}$ for the association analysis using the male sample.

**$T_{Diff}$, the interaction effect only test.** We could also consider the interaction effect only test for effect difference between females and males:

$$T_{Diff} = \frac{\hat{\beta}_F - \hat{\beta}_M}{\sqrt{\hat{v}_F^2 + \hat{v}_M^2}}. \tag{1}$$

Under the null of no effect differences between females and males, $T_{Diff}$ asymptotically follows the $N(0, 1)$ distribution.

**$T_{1,metaL}$, the traditional meta-analysis.** $T_{1,metaL}$ is the traditional meta analysis, which is an inverse-variance weighted average of $T_{Female}$ and $T_{Male}$,

$$T_{1,metaL} = \frac{1/\hat{v}_F}{\sqrt{1/\hat{v}_F^2 + 1/\hat{v}_M^2}} T_{Female} + \frac{1/\hat{v}_M}{\sqrt{1/\hat{v}_F^2 + 1/\hat{v}_M^2}} T_{Male}. \tag{2}$$

Under the null of no association, $T_{1,metaL}$ is asymptotically $N(0, 1)$ distributed. The subscript $_L$ reflects the fact that $T_{1,metaL}$ combines the two $Z$ statistics linearly.

**$T_{2,metaQ}$ Omnibus meta-analysis.** Instead of forming a directional test by linearly combining $T_{Female}$ and $T_{Male}$, we could construct an 2 df interaction test by calculating a quadratic sum as

$$T_{2,metaQ} = T_{Female}^2 + T_{Male}^2. \tag{3}$$

Under the null of no association, $T_{2,metaQ}$ asymptotically follows the centralized $\chi_2^2$ distribution.

**$T_{2,mega}$, SNP main and SNP×sex interaction joint analysis.** When individual data are available, we could consider the mega-analysis approaches through regression.

$$\mathbf{y} = a + b_g\mathbf{g} + b_s\mathbf{s} + b_{gs}\mathbf{g} \times \mathbf{s} + \mathbf{X}\mathbf{b}_X + N(\mathbf{0}, \sigma_\mathbf{s}^2\mathbf{I}_n), \tag{4}$$

where $\mathbf{y}$, $\mathbf{g}$ and $\mathbf{s}$ are sex-combined vectors of phenotype, genotype and sex respectively, and $\mathbf{X}$ is the covariate matrix for variables in addition to sex. Here the interaction term is included to capture the potential genetic heterogeneity between female and male. Additionally and importantly, the error model is sex-specific, $\sigma_\mathbf{s}^2$. That is, sex-specific variances, $\sigma_F^2$ and $\sigma_M^2$, are assigned to female and male groups, respectively; for example, human height differs between sexes in both its mean and variances [52]. We use $T_{2,mega}$ to denote the 2 df Wald test statistic derived from jointly testing $H_0 : b_g = b_{gs} = 0$. Aschard et al [27] concluded that $T_{2,mega}$ are equivalent to $T_{2,metaQ}$ and could be derived from the latter when individual data are unavailable.

**$T_{1,mega}$, SNP main effect test.** For completeness, we also consider the standard approach of testing SNP main effect. We consider the following most commonly used regression model which was also implemented in Neale Lab's both-sex analysis,

$$\mathbf{y} = a' + b_g'\mathbf{g} + b_s'\mathbf{s} + \mathbf{X}\mathbf{b}_\mathbf{X}' + N(\mathbf{0}, \sigma_\mathbf{s}'^2\mathbf{I}_n), \tag{5}$$

with the modification on the error term to allow for sex-specific variance $\sigma_\mathbf{s}'^2$, as noted earlier. We use $T_{1,mega}$ to denote the Wald test statistic derived from testing $H_0 : b_g' = 0$. Zeng and Lin [32] has shown the analytical equivalence between $T_{1,mega}$ and $T_{1,metaL}$.

### Independent SNPs, loci and lead variant annotation

We conducted linkage disequilibrium (LD) clumping using PLINK 1.9's `--clump` option to extract independent SNPs from the GWAS summary statistics [53]. For each testing method and each trait, we grouped the genome-wide significant SNPs that were within a physical distance of 10Mb and had an $r^2$ value (based on 1000 Genomes Phase 3 [54] European reference panel) greater than 0.01 as outlined in [1]. The SNP with the lowest p-value within each clump was selected to form a set of independent GWAS SNPs.

To select SNPs for functional annotation, we employed FUMA (Version v1.5.4 [55]) to merge independent SNPs within 250 kb to form risk loci. Each locus was represented by the top lead SNP which has the minimum *p*-value in the locus. We then annotated the loci with the previously reported SNP-phenotype associations of these lead SNPs and their dependent SNPs within the loci ($r^2 > 0.6$) by referencing NHGRI-EBI GWAS catalog (Version e0r2022-11-29) accessed through FUMA (Version v1.5.4 [55], date accessed: 2023-05-13). Loci that exhibited no previously reported SNP associations with any traits for their lead SNPs or their dependent SNPs were highlighted as novel signals.

## Supporting information

**S1 Appendix. Empirical equivalence between mega- and meta-analysis.**
(PDF)

**S2 Appendix. Simulation study design.**
(PDF)

**S3 Appendix. Simulation results.**
(PDF)

**S4 Appendix. Comparison of number of independent SNPs.**
(PDF)

**S5 Appendix. Analysis check on MAF.**
(PDF)

**S6 Appendix. SNPs with opposite significant effects between sexes for urate on chr4.**
(PDF)

**S7 Appendix. Analytical ranges for p-values of $T_{1,metaL}$ and $T_{2,metaQ}$ in Fig 2.**
(PDF)

**S1 Data. Loci uniquely identified by $T_{2,metaQ}$ but missed by all other four 1 d.f. tests.**
(XLSX)

**S1 Fig. Pairwise scatter plots of the genome-wide significant ($p < 5 \times 10^{-8}$) associations detected by any of the five testing methods considered.** (A) includes SNPs with either sex-stratified MAF $\leq 0.05$, and (B) includes SNPs with $MAF > 0.05$ in both sex groups, across all 290 traits analyzed and after excluding the testosterone (C and D). The five association methods include $T_{Female}$ (Female-only analysis), $T_{Male}$ (Male-only analysis), $T_{Diff}$ (SNP-sex interaction-only test), $T_{1,metaL}$ (the traditional sex-combined meta-analysis), and $T_{2,metaQ}$ (the omnibus meta-analysis); see Table 1 for method details. The sex-stratified GWAS summary statistics come from the Neale lab's UK Biobank GWAS round 2, which included a cohort of up to 361,194 participants (312,102 in testosterone GWAS, 154,364 females and 157,738 males). Axes depict $-\log_{10}$ *p*-values for each pair of tests, and each hexagon's color corresponds to the count of associations falling within the $-\log_{10}$ *p*-value range defined by that region. The

$-\log_{10} p$ maximum was truncated at 200 to improve visualization. The dashed line indicates the reference main diagonal reference line.
(TIF)

**S2 Fig. Histograms of the number of associated traits across 1,113,865 SNPs associated with one or more traits, stratified by the five testing methods.** 1,113,865 SNPs are associated with one or more traits identified by any of the five association testing methods: $T_{Female}$ (Female-only analysis), $T_{Male}$ (Male-only analysis), $T_{Diff}$ (SNP-sex interaction-only test), $T_{1,metaL}$ (the traditional sex-combined meta-analysis), and $T_{2,metaQ}$ (the omnibus meta-analysis). The histograms are based on $N_{j,m} = \sum_{t=1}^{290} I(p_{j,t,m} < 5 \times 10^{-8})$, where $I(\cdot)$ is an indicator function, $p_{j,t,m}$ is the association $p$-value between SNP $j$ and trait $t$ by method $m$. The y-axis ticks are on the $\log_{10}$ scale for ease of visualization.
(TIF)

**S3 Fig. Manhattan plots of the number of associated traits for each of the 1,113,865 SNPs associated with one or more traits, stratified by the five testing methods.** The 1,113,865 SNPs are associated with one or more traits identified by any of the five association testing methods: $T_{Female}$ (Female-only analysis), $T_{Male}$ (Male-only analysis), $T_{Diff}$ (SNP-sex interaction-only test), $T_{1,metaL}$ (the traditional sex-combined meta-analysis), and $T_{2,metaQ}$ (the omnibus meta-analysis). The plots are based on $N_{j,m} = \sum_{t=1}^{290} I(p_{j,t,m} < 5 \times 10^{-8})$, where $I(\cdot)$ is an indicator function, $p_{j,t,m}$ is the association $p$-value between SNP $j$ and trait $t$ by method $m$.
(TIF)

**S4 Fig. Pairwise scatter plots of the number of associated traits across the 1,113,865 SNPs among the five testing methods.** The 1,113,865 SNPs are the SNPs associated with one or more traits identified by any of the five association testing methods: $T_{Female}$ (Female-only analysis), $T_{Male}$ (Male-only analysis), $T_{Diff}$ (SNP-sex interaction-only test), $T_{1,metaL}$ (the traditional sex-combined meta-analysis), and $T_{2,metaQ}$ (the omnibus meta-analysis). The plots are based on $N_{j,m} = \sum_{t=1}^{290} I(p_{j,t,m} < 5 \times 10^{-8})$, where $I(\cdot)$ is an indicator function, $p_{j,t,m}$ is the association $p$-value between SNP $j$ and trait $t$ by method $m$.
(TIF)

**S5 Fig. Left: the bottom-right plot in Fig 1A, comparing $T_{2,metaQ}$ and $T_{1,metaL}$; Right: a zoomed-in view within the range of (0,20) to show that the $-\log10p$-values of $T_{2,metaQ}$ can be slightly smaller than those of $T_{1,metaL}$.** In the red-highlighted area (a), there are 179, 718 SNP-trait associations which were identified by $T_{2,metaQ}$ but missed by $T_{1,metaL}$, and in the blue-highlighted area (b), there are 756, 316 SNP-trait associations which were identified by $T_{1,metaL}$ but missed by $T_{2,metaQ}$.
(TIF)

**S6 Fig. Stacked Manhattan plots of genome-wide significant SNPs.** (A) 64,934 SNP-phenotype associations identified by $T_{2,metaQ}$ but missed by all other four methods across all the 290 traits, (B) 800,183 SNP-phenotype associations missed by $T_{2,metaQ}$ but identified by any other four methods across all the 290 traits, (C) 63,647 SNP-phenotype associations identified by $T_{2,metaQ}$ but missed by all other four methods after removing the testosterone, and (D) 797,585 SNP-phenotype associations missed by $T_{2,metaQ}$ but identified by any other four methods after removing the testosterone. The $-\log_{10}p$-values (with further $\log_{10}$ transformation on y-axis to aid presentation) are shown for the five association methods, including $T_{Female}$ (Female-only analysis), $T_{Male}$ (Male-only analysis), $T_{Diff}$ (SNP-sex interaction-only test), $T_{1,metaL}$ (the traditional sex-combined meta-analysis), and $T_{2,metaQ}$ (the omnibus meta-analysis); see Table 1 for

method details. The sex-stratified GWAS summary statistics come from the Neale lab's UK Biobank GWAS round 2, which included a cohort of up to 361,194 participants (194,174 females and 167,020 males). The red horizontal lines indicate the genome-wide significant threshold of $5 \times 10^{-8}$ on the $-log_{10}$ scale.
(TIF)

**S7 Fig. Stacked Manhattan plots of genome-wide significant SNPs with MAF greater than 0.05 in both sex groups, comparing $T_{2,metaQ}$ vs $T_{1,metaL}$.** (A) SNPs identified by $T_{2,metaQ}$ but missed by $T_{1,metaL}$ across all the 290 traits, (B) SNPs missed by $T_{2,metaQ}$ but identified by $T_{1,metaL}$ across all the 290 traits, (C) SNPs identified by $T_{2,metaQ}$ but missed by $T_{1,metaL}$ after removing the testosterone, and (D) SNPs missed by $T_{2,metaQ}$ but identified by $T_{1,metaL}$ after removing the testosterone. The $-log_{10}p$-values (with further $log_{10}$ transformation on y-axis to aid presentation) are shown for the five association methods, including $T_{Female}$ (Female-only analysis), $T_{Male}$ (Male-only analysis), $T_{Diff}$ (SNP-sex interaction-only test), $T_{1,metaL}$ (the traditional sex-combined meta-analysis), and $T_{2,metaQ}$ (the omnibus meta-analysis); see Table 1 for method details. The sex-stratified GWAS summary statistics come from the Neale lab's UK Biobank GWAS round 2, which included a cohort of up to 361,194 participants (194,174 females and 167,020 males). The red horizontal lines indicate the genome-wide significant threshold of $5 \times 10^{-8}$ on the $-log_{10}$ scale.
(TIF)

**S8 Fig. Stacked Manhattan plots of genome-wide significant SNPs with MAF greater than 0.05 in both sex groups, comparing $T_{2,metaQ}$ vs all other four methods.** (A) SNPs identified by $T_{2,metaQ}$ but missed by all other four methods across all the 290 traits, (B) SNPs missed by $T_{2,metaQ}$ but identified by any other four methods across all the 290 traits, (C) SNPs identified by $T_{2,metaQ}$ but missed by all other four methods after removing the testosterone, and (D) SNPs missed by $T_{2,metaQ}$ but identified by any other four methods after removing the testosterone. The $-log_{10}p$-values (with further $log_{10}$ transformation on y-axis to aid presentation) are shown for the five association methods, including $T_{Female}$ (Female-only analysis), $T_{Male}$ (Male-only analysis), $T_{Diff}$ (SNP-sex interaction-only test), $T_{1,metaL}$ (the traditional sex-combined meta-analysis), and $T_{2,metaQ}$ (the omnibus meta-analysis); see Table 1 for method details. The sex-stratified GWAS summary statistics come from the Neale lab's UK Biobank GWAS round 2, which included a cohort of up to 361,194 participants (194,174 females and 167,020 males). The red horizontal lines indicate the genome-wide significant threshold of $5 \times 10^{-8}$ on the $-log_{10}$ scale.
(TIF)

**S9 Fig. Stacked Manhattan plots of genome-wide significant SNPs with either sex-stratified MAF $\leq$ 0.05, comparing $T_{2,metaQ}$ vs $T_{1,metaL}$.** (A) SNPs identified by $T_{2,metaQ}$ but missed by $T_{1,metaL}$ across all the 290 traits, (B) SNPs missed by $T_{2,metaQ}$ but identified by $T_{1,metaL}$ across all the 290 traits, (C) SNPs identified by $T_{2,metaQ}$ but missed by $T_{1,metaL}$ after removing the testosterone, and (D) SNPs missed by $T_{2,metaQ}$ but identified by $T_{1,metaL}$ after removing the testosterone. The $-log_{10}p$-values (with further $log_{10}$ transformation on y-axis to aid presentation) are shown for the five association methods, including $T_{Female}$ (Female-only analysis), $T_{Male}$ (Male-only analysis), $T_{Diff}$ (SNP-sex interaction-only test), $T_{1,metaL}$ (the traditional sex-combined meta-analysis), and $T_{2,metaQ}$ (the omnibus meta-analysis); see Table 1 for method details. The sex-stratified GWAS summary statistics come from the Neale lab's UK Biobank GWAS round 2, which included a cohort of up to 361,194 participants (194,174 females and 167,020 males). The red horizontal lines indicate the genome-wide significant threshold of $5 \times 10^{-8}$ on the

$-log_{10}$ scale.
(TIF)

**S10 Fig. Stacked Manhattan plots of genome-wide significant SNPs with either sex-stratified MAF $\leq$ 0.05, comparing $T_{2,metaQ}$ vs all other four methods.** (A) SNPs identified by $T_{2,metaQ}$ but missed by all other four methods across all the 290 traits, (B) SNPs missed by $T_{2,metaQ}$ but identified by any other four methods across all the 290 traits, (C) SNPs identified by $T_{2,metaQ}$ but missed by all other four methods after removing the testosterone, and (D) SNPs missed by $T_{2,metaQ}$ but identified by any other four methods after removing the testosterone. The $-log_{10}p$-values (with further $log_{10}$ transformation on y-axis to aid presentation) are shown for the five association methods, including $T_{Female}$ (Female-only analysis), $T_{Male}$ (Male-only analysis), $T_{Diff}$ (SNP-sex interaction-only test), $T_{1,metaL}$ (the traditional sex-combined meta-analysis), and $T_{2,metaQ}$ (the omnibus meta-analysis); see Table 1 for method details. The sex-stratified GWAS summary statistics come from the Neale lab's UK Biobank GWAS round 2, which included a cohort of up to 361,194 participants (194,174 females and 167,020 males). The red horizontal lines indicate the genome-wide significant threshold of $5 \times 10^{-8}$ on the $-log_{10}$ scale.
(TIF)

**S11 Fig. Comparison of the numbers of genome-wide significant SNPs (a) and independent loci (b) identified by the traditional meta-analysis $T_{1,metaL}$ (y-axis) and the omnibus meta-analysis $T_{2,metaQ}$ (x-axis) for each of the 290 traits analyzed.** Both x-axis and y-axis are in $log_{10}$ scale, which provides a zoomed-in look at the traits with smaller numbers of associated SNPs. In (b) linkage disequilibrium is defined with a physical distance of 100kb (Methods and material). The dashed line indicates the reference main diagonal line. The plot with the original scale is provided in Fig 3.
(TIF)

**S12 Fig. Comparison of the numbers of independent genome-wide significant ($p < 5 \times 10^{-8}$) SNPs identified by traditional meta-analysis $T_{1,metaL}$ and omnibus meta-analysis $T_{2,metaQ}$ across 290 traits.** In plots A and B, the x-axis and y-axis represent the numbers of independent SNPs identified by $T_{2,metaQ}$ and $T_{1,metaL}$, respectively. The linkage disequilibrium is defined with a physical distance of 10MB and $r^2 > 0.01$. The definition for independent SNPs is available in the Materials and methods. Plot B provides a zoom-in view of plot A within the range of (0, 250). Traits that yield more signals in $T_{2,metaQ}$ than in $T_{1,metaL}$ and present at least 50 independent genome-wide significant SNPs in both methods are highlighted in red. Traits with the largest number of associations are annotated. The dashed line indicates the reference main diagonal line.
(TIF)

**S13 Fig. Histogram of the number of associated loci uniquely identified by $T_{2,metaQ}$ but missed by any of the other four methods across phenotypes.** In total, 164 out of the 290 tested phenotypes have one or more associated loci identified by $T_{2,metaQ}$ (the omnibus meta-analysis), but missed by all four of $T_{Female}$ (Female-only analysis), $T_{Male}$ (Male-only analysis), $T_{Diff}$ (SNP-sex interaction-only test) and $T_{1,metaL}$ (the traditional sex-combined meta-analysis).
(TIF)

**S14 Fig. Stacked Manhattan plots (A) and pairwise scatter plots (B) of genetic association for testosterone level.** In (A) the $-log_{10}$ $p$-values (with further $log_{10}$ transformation on y-axis to aid presentation) are shown for the five association methods, including $T_{Female}$ (Female-only analysis), $T_{Male}$ (Male-only analysis), $T_{Diff}$ (SNP-sex interaction-only test), $T_{1,metaL}$ (the

traditional sex-combined meta-analysis), and $T_{2,metaQ}$ (the omnibus meta-analysis); see Table 1 for method details. The sex-stratified GWAS summary statistics come from the Neale lab's UK Biobank GWAS round 2, which included a cohort of up to 361,194 participants (312,102 in testosterone GWAS, 154,364 females and 157,738 males). The red horizontal lines indicate the genome-wide significant threshold of $5 \times 10^{-8}$ on the $-log_{10}$ scale. In (B), axes depict $-\log_{10}$ p-values for each pair of tests. For simplicity in computation and better visualization, we only included points that achieved genome-wide significance in at least one of the five tests. The $-\log_{10} p$ maximum was truncated at 200 to improve visualization. The dashed line indicates the reference main diagonal line.
(TIF)

**S15 Fig. Sex-specific Z-scores of X chromosome SNP rs5934505 in GWAS of 290 continuous phenotypes.** The sex-stratified GWAS summary statistics come from the Neale lab's UK Biobank GWAS round 2, which included a cohort of up to 361,194 participants (194,174 females and 167,020 males). The dashed lines indicate critical values corresponding to the two-tailed test at genome-wide significant level (5E-8).
(TIF)

**S16 Fig. Histograms of numbers of genome-wide significant SNPs by tests across 290 traits.** We compare the signals identified by $T_{Female}$: Female-only analysis, $T_{Male}$: Male-only analysis, $T_{Diff}$: SNP-sex interaction-only test, $T_{1,metaL}$: Traditional sex-combined meta-analysis, and $T_{2,metaQ}$: Omnibus meta-analysis. The sex-stratified GWAS summary statistics come from the Neale lab's UK Biobank GWAS round 2, which included a cohort of 361,194 participants (194,174 females and 167,020 males). We excluded phenotypes with no signals in any of the five methods.
(TIF)

**S1 Table. 16 testosterone-associated loci uniquely identified by the recommended $T_{2,metaQ}$ but missed by any other methods in the UK Biobank data with previously reported associations in NHGRI-EBI GWAS catalog.** The $T_{2,metaQ}$ method uniquely identified 1,287 genome-wide significant SNPs, representing 36 independent loci after Linkage Disequilibrium (LD) clumping (Materials and methods). This table presents 16 of these 36 loci with leading SNPs or their dependent SNPs that have been previously reported to be associated either with testosterone phenotypes (A) or with any other phenotypes (B), as reported in the NHGRI-EBI GWAS catalog (Version: e0r2022-11-29, [38]). The $\beta_{Female}$ and $\beta_{Male}$ columns show the sex-specific effect size estimates from the stratified analysis, indicating the estimated effect of each copy of the minor allele. We compared the p-values for the following methods: $T_{Female}$: Female-only analysis, $T_{Male}$: Male-only analysis, $T_{Diff}$: SNP-sex interaction-only test, $T_{1,metaL}$: Traditional sex-combined meta-analysis, and $T_{2,metaQ}$: Omnibus sex-combined interaction meta-analysis.
(PDF)

**S2 Table. 8 novel testosterone-associated loci with leading insertions/deletions (INDELs) uniquely identified by the recommended $T_{2,metaQ}$ but missed by any other methods in the UK Biobank data.** The $T_{2,metaQ}$ method uniquely identified 1,287 genome-wide significant SNPs, corresponding to 36 independent loci after LD clumping. This table presents 8 of these 36 loci that have never been reported to be associated with any phenotypes in the NHGRI-EBI GWAScatalog [38]. Each locus is represented by its Lead SNP, selected based on the minimum p-value within the locus. The table only included loci with leading insertions/deletions (INDELs). Loci with bi-allelic leading SNPs are reported in Table 2. The $\beta_{Female}$ and $\beta_{Male}$ columns show the sex-specific effect size estimates from the stratified

analysis, indicating the estimated effect of each copy of the minor allele. We calculated the p-values for the following methods: Female-only $T_{Female}$, male-only $T_{Male}$, traditional sex-combined meta-analysis $T_{1,metaL}$, and SNP-sex interaction-only analysis $T_{Diff}$, omnibus meta-analysis $T_{2,metaQ}$. The sex-stratified summary statistics used were from Neale's group (Online Resources).
(PDF)

**S3 Table. 19 bi-allelic SNPs exhibiting genome-wide significant effects on urate in opposite directions in females and males (including IRNT results).** We compared the p-values for the following methods: $T_{Female}$: Female-only analysis, $T_{Male}$: Male-only analysis, $T_{Diff}$: SNP-sex interaction-only test, $T_{1,metaL}$: Traditional sex-combined meta-analysis, and $T_{2,metaQ}$: Omnibus meta-analysis. The $\beta_{Female}$ and $\beta_{Male}$ columns show the sex-specific effect size estimates from the stratified analysis, indicating the estimated effect of each copy of the minor allele. The sex-stratified GWAS summary statistics come from the Neale lab's UK Biobank GWAS round 2, which included a cohort of 361,194 participants (343,836 in urate GWAS, 184,755 females and 159,081 males). Columns notated with IRNT show result based on inverse normal transformed urate phenotype.
(PDF)

## Acknowledgments

We thank Dr. Shelley Bull and Dr. Lisa Strug for their helpful comments and discussion. BL is a trainee of the CANSSI-Ontario STAGE (Strategic Training for Advanced Genetic Epidemiology) training program at the University of Toronto.

## Online resources

Neale Lab's round 2 (imputed-v3) UKB GWAS summary statistics: http://www.nealelab.is/uk-biobank

Trait-specific and sex-stratified sample sizes of the UKB GWAS: https://sex-combined-interaction-meta.notion.site/3b5a7204dcb34d1eb0f4895297012e2c?v=c342e67e2fe6427abcb9b407088a48b4&pvs=4.

## Author Contributions

**Conceptualization:** Lei Sun.

**Formal analysis:** Boxi Lin.

**Funding acquisition:** Andrew D. Paterson, Lei Sun.

**Investigation:** Boxi Lin.

**Methodology:** Boxi Lin, Lei Sun.

**Supervision:** Andrew D. Paterson, Lei Sun.

**Visualization:** Boxi Lin.

**Writing – original draft:** Boxi Lin.

**Writing – review & editing:** Andrew D. Paterson, Lei Sun.

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
