## [Decision Letter · Decision Letter 0]

23 Jan 2024

Dear Dr Sun,

Thank you very much for submitting your Research Article entitled 'Better together against genetic heterogeneity: a sex-combined joint main and interaction analysis of 290 quantitative traits in the UK Biobank' to PLOS Genetics.

The manuscript was fully evaluated at the editorial level and by independent peer reviewers. The reviewers appreciated the attention to an important topic but identified some concerns that we ask you address in a revised manuscript.

We therefore ask you to modify the manuscript according to the review recommendations. Your revisions should address the specific points made by each reviewer.

Yours sincerely,

Heather J Cordell

Academic Editor

PLOS Genetics

Xiaofeng Zhu

Section Editor

PLOS Genetics

Reviewer's Responses to Questions

**Comments to the Authors:**

Reviewer #1: The manuscript titled "Better together against genetic heterogeneity: a sex-combined joint main and interaction analysis of 290 quantitative traits in the UK Biobank" compared sex-stratified and combined tests across 290 quantitative traits from UK Biobank data which have not been done. The study revealed that the omnibus meta-analysis outperformed other methods, particularly in scenarios involving heterogeneity in data. It also demonstrated comparable performance to sex-combined main effect with sex interaction tests in simulated data. The authors further illustrated the potency of the omnibus meta-analysis for traits like testosterone.

The work has been well-executed, and the introduction is particularly well-written. However, some minor changes are required.

1. Out of 13 million SNPs, 6 million show significance across 290 traits. How about examining the significant SNPs that are common across multiple traits identified by these methods? Was there any difference observed in this regard?

2. Page 5. The disparity in SNP numbers is significantly larger, with 179,718 SNPs missed by T1,metaL but identified by T2,metaQ and 756,316 SNPs missed by T2,metaQ but identified by T1,metaL across all 346 traits. Please include this information in the results section, not only in the figure legends.

3. In Fig 2A and 2B, it is noteworthy that the p-values for T1MetaL never appear to exceed 8-9, particularly when identified solely by T1MetaL. Is there any specific reason for this observed pattern or I misinterpreted from the figure?

4. If possible, could you consider adding additional axis ticks 8 to the y-axis for clarity purpose?

5. In the Discussion section on pages 10 (278-285), the additional results presented in the appendix should be integrated into the main paper. Given that there isn't any restriction on word limit, their inclusion would enhance the main paper by providing additional evidence of the method's effectiveness.

6. There are two figures with S35. Please check.

7. Page 10 line 313, Check the format “Fig S28” not “S28 Fig”

8. Check citation number 12

Reviewer #2: The manuscript reads well and the message is clear. All observations from simulation and real data analyses appear to be in agreement with the expected from the theory, and I do not have any concern with the methodology. Overall, the study does not bring new information per se, as the advantages of the joint test described in this study have already been described in multiple previous publications. Joint test meta-analysis is actually commonly applied in many real data applications investigating gene-environment interaction. However, it provides an interesting large-scale real data validation of these results. Still, some points need to be clarified. My comments are as follows:

Major comments:

1. Results, line 159-160: “it is clear that T1,metaL generally resulted in more associated SNPs than T2,metaQ”. Results from Figure 3 seem to be in contradiction with Figure 1 (line 130-131 “the −log10p-values of T2,metaQ appear to be consistently larger than those of the other four methods”). I cannot reconcile the two results: i) Fig1A-B lower right panels show that −log10p of T2,metaQ across all variants and all phenotypes are larger than those from T1,metaL. In that plot, I cannot see a single point above the diagonal. ii) Figure 3 shows that the number of significant signals from T1,metaL is almost always larger than the number of significant signals from T2,metaQ. I might be missing something but I cannot see how this is possible.

2. Results, line 174 “29,327 and 5,804”. Primary results are expressed in count of total associated variants. This is not much informative because of LD, and results per independent loci should be provided at the same time. Results per loci are currently provided only later on for specific cases (e.g. for testosterone, line 186-195).

3. There is something weird in Table 2: effect sizes for women are substantially smaller than for male. For example, for rs117599084, p-value are similar (8.15e-05 and 1.91e-05), but beta for women is an order of magnitude smaller (0.018 and 0.114). Given the MAF are similar across sex, the only explanation I can think of is that the beta standard errors are smaller for women. How could that be assuming that the sample size for male and female is the same?

Minor comments

4. Introduction, line 37-38: “its implementation requires the between-study variance estimate”. That would be the case in the presence of sample overlap only, correct?

5. Introduction, line 56-58 and further on in the manuscript (e.g. line 94, 165, 183…). “…including interaction effects makes the association analysis more robust to model assumptions in that the test would be considerably more powerful”. The term “robust” seems inappropriate and might be confusing. Robustness typically refers to the ability of a statistics to maintain its properties when underlying assumptions are violated. I do not think heterogeneity of genetic signal matches this definition. Instead, the interaction model simply improves the fit to the data, resulting in a gain of power.

6. Introduction, line 85-88. “Through a large-scale application, our study aims to provide empirical evidence to show that T2, metaQ is generally comparable to all other GWAS methods and can be more powerful in the presence of genetic effect heterogeneity.” This is correct, but I think it would be worth mentioning that stratified analyses remain of high interest if heterogeneity is suspected --simply to determine where the signal is coming from, but also for secondary analyses such as genetic correlation, or functional annotation enrichment analyses.

7. Results, line 128: “Fig 1A shows the pairwise P-P plots”, and further on in the manuscript. “P-P plot” refers to probability-probability plots, while Figure 1 presents -log10(P) plots, more commonly described in the literature using the broader definition of quantile-quantile plot (i.e. QQ-plot).

8. Results, line 147-153: it looks like that the gap between T1metaL and T2metaQ observed in real data perfectly matches the expected cost of having an additional degree of freedom in the absence of an interaction effect (i.e. when the chi-square from the interaction term is null). In short, 8.17 corresponds to a p-value of 6.76e-09 and a 1df chi-square of 33.6. The p-value from a 2df chi-square of 33.6 equals 5e-8, with -log10(5e-8) = 7.3. On the other hand, the -log10(P) = 6.45 from a 2df chi-square corresponds to the expected 5e-8 from a 1df chi-square. This can be easily derived in R using the formula below:

pchisq(qchisq(10^-8.17,1, lower.tail=F),2,lower.tail=F)

pchisq(qchisq(10^-6.45,2, lower.tail=F),1,lower.tail=F)

This might be worth mentioning along these observed results.

Reviewer #3: Lin and colleagues compared various approaches to identify genetic associations based on sex-stratified genome-wide association studies (GWAS). The approaches include female-specific analysis (testing for associations among women-only at alpha_f=5e-8), male-specific analysis (alpha_m=5e-8), difference analysis (testing for the difference between male and female genetic effects, alpha_diff=5e-8), fixed-effect meta-analysis to obtain sex-combined association (“metaL approach”, alpha=5e-8) and an omnibus test (testing for genetic main or GxSex interaction, “metaQ approach”, alpha_2df=5e-8). The omnibus test statistic is the sum of female- and male-specific chi-squared statistics and follows a 2df chi-square distribution under the null. This test is equivalent to the “2DF joint test” that is well-known in GxE literature and that that can be derived from a regression model that includes G, SEX and GxSEX interaction. The introduction cites the relevant literature on the topic. They compare the approaches using sex-stratified GWAS summary statistics from the UK Biobank (UKB) for 290 quantitative traits (~190K females, 165K males) that had previously been provided by the Neale lab (the authors did not conduct any GWAS). Confirmatory with the literature on the 2DF joint test, the authors find better power, i.e., more associations, by the metaQ approach when there is some underlying dependency of the genetic effects on sex. Without sex dependency, the metaL approach finds more associations due to the decreased degrees-of-freedom. The key point is that metaQ finds any type of association (sex dependent or not) with adequate power whereas the metaL approach only finds those without sex dependency but misses the loci that are dependent on sex. The authors thus advocate usage of the joint test, which is reasonable. This observation is not novel but confirmatory to what is known from the literature. However, this is the first report to apply the joint test to the Neale data set on a large scale. The authors revealed some novel associations for specific traits that have been missed previously. These are of interest for researchers studying the respective traits.

While I think that the paper is well done and robust, I still have some minor comments:

1. How is relatedness between males and females accounted for in their tests, particularly in the omnibus test? Is this possible at all or can it just be applied to unrelated males/females? I missed information on whether males/females in this analysis include relatives or not.

2. I am missing references to sex-stratified GWAS of body fat distribution that have identified multiple sex-specific loci (including women-only, male-only and even opposite effects).

3. The authors highlight associated loci for testosterone (36 loci) and serum urate (1 locus) in the main text and main tables. Have there not been other loci for other traits that could only be identified by the joint test? Any other results are only shown in figures in terms of P values and it is difficult to find “novel” loci across traits. Can the authors provide an overview of identified loci by trait and test and show the index variants of the sex-specific associations for any trait?

4. 36 independent loci have been identified for testosterone. These include 20 that have not been identified for any other trait before (when compared to GWAScatalog). Table 2 shows 12 of the 20 that were bi-allelic. Table S2 shows the other 8. These are termed “multiallelic SNPs”. In fact these are insertions/deletions (INDELs) and no SNPs. A multiallelic SNP has multiple base exchanges at a single position (eg

---

## [Decision Letter · Decision Letter 1]

11 Mar 2024

Dear Dr Sun,

We are pleased to inform you that your manuscript entitled "Better together against genetic heterogeneity: a sex-combined joint main and interaction analysis of 290 quantitative traits in the UK Biobank" has been editorially accepted for publication in PLOS Genetics. Congratulations!

Yours sincerely,

Heather J Cordell

Academic Editor

PLOS Genetics

Xiaofeng Zhu

Section Editor

PLOS Genetics

Comments from the reviewers (if applicable):

Reviewer's Responses to Questions

**Comments to the Authors:**

Reviewer #1: The authors have diligently addressed all the comments provided, taking into consideration feedback from other reviewers as well. This collaborative effort has resulted in improvements in the clarity of the manuscript. I have no further suggestions for revision.

Reviewer #2: I thank the authors for carefully addressing all my questions. I do not have further comments.

Reviewer #3: The authors have adressed all of my comments.

**Have all data underlying the figures and results presented in the manuscript been provided?**

Reviewer #1: Yes

Reviewer #2: Yes

Reviewer #3: Yes

PLOS authors have the option to publish the peer review history of their article (what does this mean?). If published, this will include your full peer review and any attached files.

Reviewer #1: No

Reviewer #2: No

Reviewer #3: No

**Data Deposition**

http://datadryad.org/submit?journalID=pgenetics&manu=PGENETICS-D-23-01346R1

**Press Queries**

---

## [Editor Report · Acceptance letter]

18 Apr 2024

PGENETICS-D-23-01346R1 

Better together against genetic heterogeneity: a sex-combined joint main and interaction analysis of 290 quantitative traits in the UK Biobank 

Dear Dr Sun, 

We are pleased to inform you that your manuscript entitled "Better together against genetic heterogeneity: a sex-combined joint main and interaction analysis of 290 quantitative traits in the UK Biobank" has been formally accepted for publication in PLOS Genetics! Your manuscript is now with our production department and you will be notified of the publication date in due course.

With kind regards,

Bernadett Koltai

PLOS Genetics

On behalf of:
